

# QUAL-NET, a high temporal resolution eutrophication model in large hydrographic networks

Camille Minaudo[1,2], Florence Curie[1], Yann Jullian[3], Nathalie Gassama[1], Florentina Moatar[1]

[1]E.A. 6293 GeHCO, François Rabelais de Tours University, Tours, 37000, France

[2]OSUR-CNRS, Rennes 1 University, Rennes, 35000, France

[3]CaSciModOT, UFR Sciences et Techniques, François Rabelais de Tours University, Tours, 37000, France

*Correspondence to*: Camille Minaudo (camille.minaudo@univ-tours.fr)

**Abstract.** To allow climate change impact assessment on river system water quality, the scientific community lacks efficient deterministic models able to simulate hydrological and biogeochemical processes in drainage networks at the regional scale,

with a fine temporal resolution and with water temperature explicitly determined. The model QUALity-NETwork (QUAL-NET) was developed and tested on the Middle Loire River Corridor, a sub-catchment of the Loire River (France), prone to eutrophication. Hourly variations computed by the model helped disentangle the complex interactions existing between hydrological and biological processes across different timescales. Phytoplankton variations in the Loire River were governed by phosphorus availability and transit time. Model QUAL-NET showed that a large amount of phytoplankton cells growing

in the upper part of the studied corridor was recycled through the microbial loop, which enhanced phytoplankton growth, explaining why severe blooms still occur in the Loire River despite large P input reductions.

## 1 Introduction

River eutrophication has become a rising problem over the past decades, especially in India, Asia or South America, constituting a major risk for ecosystems and human health (e.g. Braga et al., 2000; Dixit et al., 2017; Némery and Garnier,

2016; Yin et al., 2016). Significant efforts to reduce non-point and point sources of nitrogen (N) and phosphorus (P) were done in Europe and North America, leading to eutrophication decline in several large rivers (Friedrich and Pohlmann, 2009; Hardenbicker et al., 2014; Hartmann et al., 2007; Howden et al., 2010; Minaudo et al., 2015, 2016). Yet, eutrophication crises are still occurring in many freshwater areas.

Previous studies often tried to assess which controlling factor of eutrophication prevails over the others, and often opposed

nutrients availability to supposedly favorable physical conditions. Conflicting results shown in the literature did not help solve this issue. Indeed, in some rivers chlorophyll-*a* concentration could directly be assessed confidently from P concentration (e.g.



Basu and Pick, 1996; Dodds, 2006), whereas river flow conditions in other systems clearly constrained and determined the algal biomass (Biggs and Smith, 2002; Istvánovics et al., 2009). A few studies identified a combination of variables co-controlling phytoplankton blooms like the association of river flow conditions, water temperature and sunshine duration over the preceding days (Bowes et al., 2016), flow and light intensity (Hardenbicker et al., 2014), and flow, temperature and

5 nutrients availability (Van Vliet and Zwolsman, 2008). If reducing P inputs has proved to be efficient to limit phytoplankton blooms in rivers, many recent studies show that both N and P availability must be considered as key elements to determine trophic state in streams and rivers (Dodds and Smith, 2016; Paerl et al., 2016). Apart from nutrients availability, numerous other factors control phytoplankton composition and abundance in rivers, such as water residence time (directly linked to the river morphology, with potential presence of flow velocity dead-zones), penetration of solar radiation into the water column

(depth and turbidity), water temperature variations (hydrological and climate forcing), invertebrate grazing from endemic and invasive species and self-shading effects by the phytoplankton colony itself (Reynolds, 2006; Reynolds and Descy, 1996).

Disentangling the relative influence of so many chemical, biological and physical factors on the river biogeochemistry can hardly be captured confidently through a simple water quality monitoring and often requires the help of numerical modelling. Many deterministic water quality models at the catchment scale were developed and used initially to estimate nutrient source

inputs into receiving waterbodies, and support watershed stakeholders and decision-makers to tackle eutrophication issues (Wellen et al., 2015). Yet, a limited number of models propose a mechanistic module simulating phytoplankton community dynamics and its impact on eutrophication. One can cite RIVERSTRAHLER (Billen et al., 1994; Garnier et al., 2002), ProSe (Even, 1995; Even et al., 1998; Flipo et al., 2004; Vilmin et al., 2015), PEGASE (Deliège et al., 2009), QSIM (Kirchesch and Schöl, 1999; Schöl et al., 1999), WaterRAT (McIntyre and Wheater, 2004), QUAL2KW (Pelletier et al., 2006), WASP7

(Ambrose and Wool, 2009), QUASAR (Whitehead et al., 1997) or RWQM1 (Reichert et al., 2001). However, many of these models are only able to simulate river stretches, and not the entire river network. The main reason is that very few models work at the catchment scale with a subdaily timestep (Wellen et al., 2015), mostly because program developers have to face long calculation time, and usually make a compromise between large spatial scale and high temporal and/or spatial resolution. Yet, the use of a fine temporal resolution is required to account for hydrological and biogeochemical processes occurring over

short periods of time (e.g. storm events, or subdaily phytoplankton growth variations). Additionally, water temperature is a key factor for phytoplankton abundance and assemblage (Reynolds, 2006) which needs to be simulated at high temporal frequency to assess the impact of potentially drier streams and warmer summers under climate change (Quiel et al., 2010). Developing methods appropriate to the regional scale is also required to account for instream processes in large rivers which control N, P and carbon (C) variations, and constrain water quality in estuarine and coastal zones. Additionally, models' scales

also have to match with the scale of actions undertaken by water body stakeholders and catchment managers.

In this context, the model QUALity-NETwork (QUAL-NET) was developed to simulate hydrological and biogeochemical processes in drainage networks at the regional scale (over $10^4$ km²), with a sub-daily temporal resolution and water temperature



explicitly determined to allow potential climate change impact assessment. QUAL-NET was based on the integration of a biogeochemical model, Rɪᴠᴇ (Garnier et al., 2002) in a thermal model, T-NET (Beaufort et al., 2016). This approach was tested on a selected portion of the Loire River basin, the Middle Loire River Corridor, draining 43 10³ km², where the river main stem (270 km long) is prone to eutrophication in summer (Lair and Reyes-Marchant, 1997; Minaudo, 2015; Minaudo et al., 2015).

## 2 Study site

The Loire River (110 10³ km²) is the largest river flowing in France. The selected Middle Loire Corridor is a subcatchment located in the lowland section of the river main stem (Figure 1). It separates the Upper Loire (a mountainous area where anthropogenic pressures are highly impacting the river water quality but where eutrophication is only visible in lakes and reservoirs), from the Lower Loire River where the river main stem meets its major tributaries (Cher, Indre, Vienne, Maine). The Middle Loire River Corridor (MLRC) starts 450 km from the source of the River and runs over 300 km, increasing in length by 50% while meeting only minor tributaries. From the beginning of the MLRC to its outlet, i.e. stations S1 to S2 respectively on Figure 1, the cumulated catchment area increases by only 26%. The MLRC has a high eutrophication potential, combining most of the conditions favoring phytoplankton growth: low water level in summer (≈1m) and a river morphology with multiple channels and numerous islands slowing down flow velocity which increases the water travel time (Latapie et al., 2014). Many efforts have been conducted since 1990 to limit phosphorus point and non-point sources and counteract eutrophication which was reaching some extreme levels (chlorophyll-*a* concentration often over 250 µg L⁻¹ in the 1980s). Phosphorus concentrations were since divided 2.5-fold and phytoplankton blooms declined 3-fold (Floury et al., 2012; Minaudo et al., 2015; Oudin et al., 2009). Yet, algal blooms are still occurring from time to time, questioning the source of phosphorus.

## 3 Methods

The model QUALity NETwork (QUAL-NET) was developed based on a deterministic approach. It is the fusion between a thermal model T-NET (Beaufort et al., 2016), and a biogeochemical model, Rɪᴠᴇ (Garnier et al., 2002).

Model T-NET is a physically based model able to estimate the water temperature in each reach of a large hydrographical network (10⁵ km²) with an hourly resolution (Beaufort, 2015; Beaufort et al., 2016). It has previously been developed specifically for the Loire River Basin (110 10³ km² and over 50 10³ river reaches from headwaters to the estuary). The temperature in the river network is computed as follows: i) resolution of the heat budget in a given reach and estimation of the equilibrium temperature; ii) longitudinal propagation downstream of the thermal signal according to the estimated water velocity throughout the river reach; iii) discharge-weighted mix of the thermal signal when two or more streams meet in one node.

Model RIVE is a mechanistic model describing many of the biogeochemical interactions occurring in the river between the water column, and the benthos. It simulates the dynamic of dissolved and particulate organic matter, nutrients (N, P, Si), dissolved oxygen, the phytoplankton biomass (three algae groups: green algae, diatoms, and cyanobacteria), zooplankton and bacteria. RIVE is the core of RIVERSTRAHLER (Billen et al., 1994) and ProSe (Even et al., 1998) models. RIVERSTRAHLER was

largely used in past studies to simulate with a 10-day time step the biogeochemical functioning of large lowland eutrophic rivers under varying climate conditions, e.g. the Seine basin, the Danube River, the Red River in Vietnam, and over large periods of time (Billen et al., 2001; Billen and Garnier, 2000; Garnier et al., 1995, 2002, 2005; Quynh et al., 2010). The numerous variables included in the model and equations are extensively described in Billen et al. (1994) and Garnier et al. (2002). Both the water and the benthic components are considered, including chemical and physical exchanges in-between

these two components, according to Billen et al. (2014) formulation.

Temporal resolution of QUAL-NET is hourly even if biogeochemical variations were computed every 15 minutes to avoid potential numerical drifts. QUAL-NET was coded in C++ language and allowed parallel computing, i.e. the simultaneous use of several processors in order to reduce as much as possible computation time.

### 3.1 Data inputs and main spatialization choices

Hydrological, geomorphological and meteorological forcing variables were determined and used on the basis of T-NET model implementation on the Loire Basin (Figure 2). Thus, a more detailed description is available in Beaufort et al. (2016), except for nutrient sources forcing variables.

### 3.1.1 Meteorological variables

Hourly meteorological variables were taken from SAFRAN atmospheric reanalysis (Quintana-Segui et al., 2008), produced

by the French Meteorological Services (Meteo-France). Spatial resolution was 8x8 km². Meteorological variables were used to compute the hydrological model (see below) for both thermal and biogeochemical modules: air temperature, specific humidity, wind velocity and atmospheric radiation were used to compute water temperature; most biogeochemical variables were water-temperature dependent, and phytoplankton photosynthesis processes were directly linked to atmospheric radiation variations.

### 3.1.2 Hydrology

Daily mean discharge and groundwater flows were simulated by the semi-distributed hydrological model EROS (Thiéry and Moutzopoulos, 1995) at the outlet of 17 subwatersheds. Within each of these subwatersheds, flows were redistributed into the hydrographic network according to the corresponding drainage area of each river reach. This approach proved its efficiency



and reliability at the regional scale in the Loire Basin (Beaufort et al., 2016). Discharge and groundwater flows were considered constant over 24 h even if the water quality model output was hourly.

### 3.1.3 Geomorphology

The hydrographical network was determined from the Carthage® database (French Ministry of Environment and regional
water agencies cartography, Carthage, 2012), after transforming multiple channels into single channels. In the MLRC, we counted 3361 reaches, every one of them being defined as the river section between two confluences. Slopes for each reach were assessed based on a 25 m resolution digital terrain model (BD ALTI®, 2012). Streams transversal morphology were assumed to be rectangular, while depth and width were assessed on a daily time step but differently for the Loire River main stem and other streams: i) depth in the Loire River main stem reaches was assessed based on field measurements conducted
during both low and high flow periods (Latapie, 2011; Latapie et al., 2014) and considering Manning-Strickler formulation with a Strickler coefficient to be calibrated numerically; ii) in all other rivers and streams, where no field measurements were done, depth and width were assessed daily based on the ESTIMKART application (Lamouroux et al., 2010) which uses reach slope, watershed area, daily and inter-annual discharge to estimate streams morphology.

### 3.1.4 Non-point sources

Non-point sources of nutrients and exports of TSS were defined based on land use (European Corine Land Cover dataset, 2006), climate characteristics, lithology (LITHO®, 2008) and previous observations conducted in 108 streams located in the Loire headwaters, upstream any potential point sources (Blanchard, 2007). Overall, land use categories were grouped into seven large categories (urban, arable land, cultivated land, prairie, forest, wetland, other types), and associated with a corresponding non-point source concentration for the following variables: nitrate, ammonium, total inorganic phosphorus,
biogenic silica, dissolved and particulate organic carbon for three different biodegradability classes, total suspended solids, and fecal matter. The MLRC basin was divided into 479 small sub-catchments (the average was 27 km²), and diffuse sources concentrations were applied homogenously for all streams located in a given sub-catchment as a combination of concentrations originating from all the different land use types. Land use was considered constant over time, leading to constant nutrient concentrations for non-point sources. Thus, it was hypothesized that the hydrological variability alone could be responsible
for seasonal and event-based variations of non-point nutrient fluxes.

### 3.1.5 Point sources

Industrial and domestic point sources of nutrients and TSS fluxes originated from Loire Basin water authorities (AELB) surveys conducted in 2010. In the MLRC basin, 641 waste water treatment plant (WWTP) were recorded. Datasets provided total organic carbon, total nitrogen and total phosphorus fluxes for all of them. Fluxes were divided into the different chemical




forms for C, N and P, according to Servais and Billen (2007) depending on the type of point sources and the characteristics of the waste water treatment. Fluxes and concentration of point sources were considered constant over time.

### 3.1.6 Upstream boundary in the Loire River and validation dataset at catchment outlet

A daily survey was conducted at S1 (Saint-Satur-sur-Loire) and S2 (Cinq-Mars-la-Pile) in the Loire River during the period
August 2011-July 2014(Minaudo, 2015). Data collected at S1 was used as data input for the model, and data at S2 was used for both calibration and model performances assessment. Samples were collected every day from a bridge using the same procedure at each station.. Total suspended solid concentrations (TSS) were measured every day. The following parameters were analysed on a 3-day frequency basis: dissolved and particulate organic carbon (DOC and POC), total and dissolved inorganic phosphorus (TP and SRP), nitrate ($NO_3^-$), dissolved silica (Si) and chlorophyll-*a* concentrations. Filtrations were
immediately made on-site using a 0.45 µm cellulose acetate membrane filters for chemical parameters and 0.70 µm glass filter (Whatman GFF) previously burned at 500°C during 6 hours for chlorophyll-*a* and POC. Total suspended solids concentrations where determined by filtration of a precise volume of each water sample through pre-weighed filters and by drying them at 105°C. After filtration, water samples and filters were stored at -80°C in polypropylene tubes after acidification of aliquots for $NO_3^-$, SRP and DOC analysis. Tubes and filters were unfrozen on the day of the analysis. DOC concentrations were measured
with a carbon and analyzer (Shimadzu TOC-V CSH/CSN). The $NO_3^-$ concentration was determined by ionic chromatography. Phosphorus was measured by colorimetry after solid digestion (potassium-persulfate digestion) in the case of TP analysis. Dissolved silica (Si) was measured by colorimetry. For POC analyses, the filters were first treated with HCl 2N to remove carbonates, dried at 60°C for 24 hours and then measured with a C/S analyzer (LECO C-S 200). Chlorophyll-*a* was measured by fluorimetry at a wavelength > 665 nm after an excitation step between 340 and 550 nm. Chlorophyll-*a* concentrations were
expressed in mg C $L^{-1}$ considering C:Chl-a ratio equals 31 according to Minaudo et al. (2016) and constituted the variable hereafter named 'PHY'.

### 3.2 Computation steps in the model based on a network topology

Computation in the model was based on a network topology: each reach in the hydrographic network corresponded to the stream segment between two confluences. Each reach was constituted by an upstream and a downstream node (Figure 1).
Then, except for first Strahler order streams in the headwaters, upper nodes were always connected to two downstream nodes.

### 3.2.1 Initialization at upper node and boundary conditions

All variables were initialized at the upper node of first Strahler order streams. Water component variables were initialized according to non-point sources estimated for hillslope catchments upstream the upper nodes. Sediment component variables were initialized homogeneously everywhere in the stream network, based on the hypothesis that the model should quickly





modify values in the sediment component, depending on the interactions with variables from the water component. The upstream boundary in the Loire River (S1) was determined based on the daily survey conducted at S1 (see above section).

### 3.2.2 Propagation downstream

All variables computed at one reach in the water component were transferred downstream according to travel time estimated from discharge and stream morphology. Variables from the benthic component interacted with the water component but were not transferred downstream. For a given time step, a given reach was discretized depending on the estimated travel time. If travel time was less than 1 hour, the reach was not segmented: thermal and biogeochemical equations were solved at the downstream node considering all forcing variables as constant because their resolution was at best hourly. If travel time exceeded one hour, the reach was segmented into as many sub-segments as needed to get one hour travel time sub-segments. This allowed calculation with a hourly resolution, and in the latter case, thermal and biogeochemical equations were solved downstream each sub-segment considering varying forcing variables with time. Within one hour time step, all biogeochemical equations were solved with a 15 minutes sub-time step, all other variables being considered constant to avoid potential numerical resolution drifts. When two streams met, the thermal and biogeochemical signals were mixed with respect to their discharge and this determined all the variables values for the next downstream upper node. Because the exact location of the WWTP input within a segment was unknown, point sources fluxes were considered to happen at the downstream node only.

### 3.3 Calibration step

The thermal model was fully deterministic and no calibration step was needed. Despite the fact that RIVE was built as a universal representation of the mechanisms occurring in rivers, some processes were based on empirical relationships. Nearly 150 coefficients were counted overall (Fig. 3), the majority of them were used to describe bacteria and phytoplankton dynamics depending on light intensity, water temperature, and nutrient availability. Most coefficients are currently accepted as universal constants, but several studies pointed out that hydro-sedimentary and P sorption/desorption processes needed experimental or numerical calibration (Vilmin et al., 2015), especially because the processes involved highly impacted performances on phytoplankton and water quality predictions (Aissa-Grouz, 2015). Phosphorus dynamic in the water compartment was based on the Langmuir equilibrium concept (Limousin et al., 2007), a description largely found in the literature for water quality models (e.g. Chao et al., 2010; Rossi et al., 2012; Vilmin et al., 2015). Very different values for P sorption/desorption coefficients according to Langmuir equilibrium equations were found experimentally or numerically in the literature, with up to 5 orders of magnitude differences from one study to another (Vilmin et al., 2015). No specific laboratory experiments were conducted in the Loire River, leading us to deploy numerical optimization methods to calibrate TSS and SRP dynamics. Because SRP computation relies on TSS dynamic, the first variable to be calibrated was TSS. Calibration was conducted by changing the values of the different coefficients to be calibrated over a range of values found in the literature. The best set of coefficients was selected when results minimized root mean square errors (RMSE) of the calibrated variable. Among the period




of records (August 1$^{st}$ 2011 to July 31$^{st}$ 2014), the period selected for calibration was the first year, i.e. August 1$^{st}$ 2011 to July 31$^{st}$, 2012, and the remaining time series served as validation.

### 3.3.1 Calibration of TSS dynamic

Total suspended solids concentration increments ($dTSS$) were computed based on a simple difference between eroded matter

from the river bed ($eros_{TSS}$) and sedimentated particles ($sedim_{TSS}$), as described in Equ. (1-4).

$$dTSS(t) = eros_{TSS}(t) - sedim_{TSS}(t) \qquad (1)$$

$$eros_{TSS}(t) = \frac{Vs_{TSS}}{depth(t)}(Cap_{TSS}(t) - TSS(t-1))\frac{SED(t-1)-SED_0}{SED_0} \qquad (2)$$

$$Cap_{TSS}(t) = Veli_0 + Veli_1 \cdot V(t)^3 \qquad (3)$$

$$sedim_{TSS}(t) = \frac{Vs_{TSS}}{depth(t)}TSS(t-1) \qquad (4)$$

where $Vs_{TSS}$ was the sedimentation velocity; $depth(t)$ was the water depth at time $t$; $Cap_{TSS}$ was the erosion capacity depending on coefficients $Veli_0$, $Veli_1$, and flow velocity $V(t)$; $SED$ was the height of the layer of sediments potentially erodible, $SED_0$ was the layer of sediments set during initialization step.

Thus, TSS concentration depended on coefficients $Veli_0$, $Veli_1$ and $Vs_{TSS}$, and were chosen as variables for TSS dynamic calibration.

### 3.3.2 Calibration of P dynamic

The SRP concentration was estimated based on sorption/desorption equations originating from Langmuir equilibrium displayed by Equ. (5) and (6). This formulation requires the maximal sorption capacity of P onto suspended solids ($Pac$, in mg P g$^{-1}$) and a half-saturation constant ($Kpads$, in mg P L$^{-1}$) to be defined.

$$dSRP(t) = \frac{1}{2}\left[A(t)^2 + 4 \cdot TIP(t) \cdot Kpads]^{\frac{1}{2}} - A(t)\right] \qquad (5)$$

$$A(t) = Kpads - TIP(t) + TSS(t) \cdot Pac \qquad (6)$$

Where $TIP$ corresponded to total inorganic phosphorus concentration at time step $t$, $Kpads$ and $Pac$ were the two parameters needing to be calibrated.





### 3.4 Model performance criteria for validation

Bias and standard deviation errors for the entire period of validation (August 1$^{st}$ 2012 to July 31$^{st}$ 2014) were calculated for each variable observed at S2. They were also computed seasonally over the period: "summer" corresponded to the bloom season, from April to October; "winter" corresponded to remaining part of the year.

### 3.5 Lagrangian point of view and fluxes budgets

In addition to more common way of presenting results longitudinally along the main river main stem, we proposed two other graphical representations of transfers and biogeochemical transformations from S1 to S2. One representation consisted in following the same water body transferred from S1 to S2, as a Lagrangian point of view. This representation was both spatial and temporal since it displayed longitudinal variations according to travel time going downstream. It was used for two typical situations, one in winter (starting on February 9$^{th}$ 2013 during a high-flow period), and another during a phytoplankton bloom (starting on July 10$^{th}$ 2012). Additionally, seasonal average fluxes budgets of all the main processes and potential inputs occurring between S1 and S2 over "winter" or "summer" periods were computed for a selection of variables (TSS, NO$_3^-$, total inorganic P, Si, PHY, and O$_2$). In those graphs, arrow widths were proportional to the corresponding calculated flux, allowing to compare the two different seasons.

### 4 Results

### 4.1 Calibration step

The best set of coefficients that minimized errors over the period are displayed in Table 1. RMSE on calibrated variables were 15 mg L$^{-1}$ for TSS and 14 µgP L$^{-1}$ for SRP. The selected values for TSS largely differed from other values found in the literature, justifying the need for this calibration step. Compared to the Seine River, it appeared necessary to increase the erosion capacity ($Veli_1$) but also to reduce considerably suspended solids sedimentation rates ($Vs_{TSS}$), which resulted on an increased sediment reactivity in the system. Values calibrated for P sorption processes were close to the values found experimentally in the Seine basin (Aissa-Grouz, 2015).

### 4.2 Model performances at station S2

Over the study period, discharge variations at S2 presented highly seasonal variations (Fig. 4): Q ranged between 60 and 150 m$^3$ s$^{-1}$ in summer low flows, and peaked over 1200 m$^3$ s$^{-1}$ in winter high flows. Observed TSS concentrations co-varied with discharge, and ranged between nearly 0 in summer to 150 mg L$^{-1}$ during high flows. The model predicted well TSS dynamic and errors bias ± standard deviation over the entire study period (August 2011 to July 2014) were 8 ± 13 mg L$^{-1}$. A few storm



events that were observed with the daily survey were however not represented by the model, especially for several storm events that occurred during low flow periods.

Phytoplankton concentrations presented three clearly delimited bloom events, between March and September of each hydrological year surveyed. The maximum concentrations recorded were between 60 to 70 µg chl-$a$ L$^{-1}$ corresponding to 1.6

and 1.9 mgC L$^{-1}$. Phytoplankton variations simulated by the model succeeded at representing seasonal variations. Errors over the entire period were 0 ± 0.4 mg C L$^{-1}$. One event at the end of summer 2012 was simulated but this did not correspond to the observations.

Nitrate concentrations presented a clear seasonal signal, fluctuating between ≈ 1.5 mgN L$^{-1}$ in summer to ≈ 3.5 mgN L$^{-1}$ in winter. The model successfully reproduced these seasonal variations, and errors were 0.1 ± 0.4 mg N L$^{-1}$.

Recorded dissolved silica concentrations ranged between nearly 0 and 8 mg Si L$^{-1}$. Concentrations always peaked in winter during high flows, and dropped in spring, concomitantly with the start of phytoplankton activity. Errors from model were large for this element (0.2 ± 1.7 mg N L$^{-1}$), especially for winter periods.

Soluble reactive P concentrations presented a clear seasonal cycle, with very low concentrations reached during summer (< 10 µg P L$^{-1}$) and relatively high concentrations in winter (≈ 60 µg P L$^{-1}$). The model represented successfully these seasonal

variations; results were subject to -2 ± 14 µg P L$^{-1}$. Diel fluctuations in summer estimated with model QUAL-NET fluctuated between 0 and 15 µg P L$^{-1}$.

Recorded particulate organic carbon concentrations ranged between 0.4 to 5 mg C L$^{-1}$, with a strong correlation on the one hand between POC and TSS in winter, and on the other hand between POC and phytoplankton biomass during algae blooms (Minaudo et al., 2016). Errors from the model were 0.3 ± 1 mg C L$^{-1}$, especially due to POC overestimation in May 2012.

Recorded dissolved organic carbon concentrations ranged between 4 and 10 mg C L$^{-1}$. The highest concentrations were observed during high flow periods, but no clear seasonal variations could be deciphered. Model QUAL-NET provided results within the measured range of values, but errors over the entire period were 0.4 ± 1.5 mg C L$^{-1}$.

Dissolved oxygen was not measured, but concentration simulated by QUAL-NET presented a clear seasonal cycle, with high values (≈ 12 mg O$_2$ L$^{-1}$) reached during winter, and low values (6 to 9 mg O$_2$ L$^{-1}$) found in summer. During phytoplankton

blooms, simulated O$_2$ concentrations were subject to large diel fluctuations, with a minimum occurring around midnight, and a maximum reached by noon.



Model performances appeared similar between seasons (Table 2) with approximately the same range of errors, except for dissolved silica whose simulated concentrations in winter were subject to higher imprecisions (2.1 against 1.3 mgSi L$^{-1}$) and for PHY with much lower absolute errors in winter but this corresponded to very low PHY concentrations.

### 4.3 Lagrangian views of winter versus summer dynamics

5   The Lagrangian views of the evolution of the different biogeochemical species highlighted different hydro-biogeochemical functioning depending on the season, (Figure 5).

i)   The selected winter event corresponded to a high-flow period: Q at S1 was 940 m$^3$ s$^{-1}$ and increased to 1110 m$^3$ s$^{-1}$ by the time the water arrived at S2. It took almost 2 days for the water to travel between S1 and S2 ($\approx$ 250 km). Most elements were simply transferred downstream, with no significant transformation or alteration between S1 and S2. Concentration of TSS presented a decreasing evolution from 33 mg L$^{-1}$ at S1 to 25 mg L$^{-1}$ at S2. Nitrate concentration slightly increased from 2.8 to 3.1 mg N L$^{-1}$ (+11%), and so did SRP (+40%). Dissolved silica concentration decreased (-12%). Phytoplankton activity remained very low and declined steadily (5 to 2 µg chl. $a$ L$^{-1}$). Dissolved oxygen slightly increased (+8%).

ii)   During the selected summer event, discharge was much lower: Q was 330 m$^3$ s$^{-1}$ when the water left S1 on July 10$^{th}$, 2012, and increased to 340 m$^3$ s$^{-1}$ when the water reached S2. The model estimated that it took nearly 3 days for the water to cover the distance between from S1 to S2, and the biogeochemical variables were largely modified while travelling downstream. Two steps were identified.

-   The first 2.5 days, total phytoplankton concentration increased from 0.5 to 1.7 mg C L$^{-1}$. Simultaneously, SRP was dramatically depleted from 50 to nearly 0 µg P L$^{-1}$. Nitrate, silica and oxygen concentrations slightly decreased ($\approx$ -10%). The amount of P released from organic matter mineralization remained limited but reached a first peak concomitantly with a large P uptake from the phytoplankton colony. Phytoplankton mortality rates kept increasing while going downstream, and peaked when growth rate reached its maximum (0.15 mgC L$^{-1}$ h$^{-1}$) when travel time from S1 was 2.3 days.

-   Then, during the next 24 hours, i.e. the time needed for the water to reach S2, phytoplankton concentration started to decrease (-15%), SRP remained very low under 5 µg P L$^{-1}$ and presented a diurnal fluctuation with a minimum reached during the afternoon, and rising concentrations when arriving at S2 by night. During this phase, organic matter mineralization as a source of inorganic P increased substantially from 2 to 13 µg P L$^{-1}$ h$^{-1}$ and, phytoplankton growth first dropped from 0.15 to near 0 mgC L$^{-1}$ h$^{-1}$ and then rose again to 0.1 mgC L$^{-1}$ h$^{-1}$ when SRP input from mineralization counteracted phytoplankton uptake.




### 4.4 Storm event disturbance during a phytoplankton bloom

A storm event occurred in August 2013, during a phytoplankton bloom. Over five days (August 9[th] to 14[th]), discharge at S2 increased from 200 to 406 $m^3 s^{-1}$ and then declined to reach 230 $m^3 s^{-1}$ on August 19[th]. This largely disturbed TSS, SRP and PHY dynamics (Fig. 6).

This storm event entailed a suspended solids peak which propagated over the entire studied river stretch. TSS concentration peak amplitude decreased from 120 to 50 mg $L^{-1}$ while flowing downstream, and the peak width widened. At the beginning of the event, SRP concentration profile was showing a complete P depletion starting approximately 80 km downstream S1. This P limitation threshold progressively moved further downstream when the storm event hit. SRP slightly increased at S2, but concentrations remained very low. When the discharge peak hit S2 (August 14[th]), SRP concentrations presented a steady

longitudinal decline from 50 µg P $L^{-1}$ down to nearly 0. Before the storm event, phytoplankton concentrations were showing a limited longitudinal increase, from 0.5 to 1.2 mg C $L^{-1}$. When the discharge peak event hit, PHY concentrations decreased in the upper part of the Middle Loire River Corridor, but clearly increased in the lower part. Phytoplankton was flushed away by the storm event, and concentrations during discharge recession were showing an increasing longitudinal profile from 0.1 to 1.1 mg C $L^{-1}$. PHY concentrations began to increase again everywhere along S1 to S2 when hydrological conditions stabilized.

### 4.5 Fluxes, transfers and transformations in the Middle Loire River Corridor

Results were similar to the Lagrangian analysis, i.e. proportions of the different contributions or biogeochemical transformations were largely depending on the season (Figure 7).

In winter, most of the biogeochemical species entering the MLRC at S1 were transferred downstream, with non-significant interactions with the biological component. Suspended solids and particulate P showed an almost balanced budget between

erosion and sedimentation processes. Lateral contribution between S1 and S2 remained small compared to the upstream flux at S1, except for nitrate because tributaries and lateral non-point sources inputs contributed to 25% of the total $NO_3^-$ flux at S2. Reaeration of the water body represented a significant portion of dissolved oxygen budget at S2 (14 %).

In summer low flows, the biological component largely modified the river biogeochemistry in the studied sector.

i)      Nitrate fluxes were 15% higher at S2 (38 t N $day^{-1}$) than at S1 (28 t N $day^{-1}$) despite N uptake by phytoplankton

(3.2 t N $day^{-1} \approx$ 11% S1 flux) and a moderate contribution from the lateral streams (12 t N $day^{-1}$). Diffuse sources in the tributaries corresponded to 94% of lateral inputs.

ii)     Inorganic phosphorus loads were divided 3-fold between S1 and S2 (from 1 t P $day^{-1}$ to 0.3 t P $day^{-1}$) due to phytoplankton and bacteria uptakes (respectively 2.6 and 0.4 t P $day^{-1}$). Interestingly, P recycling from organic matter mineralization (phytoplankton dead cells) supplied 1.3 t P $day^{-1}$, i.e. more available phosphorus than





upstream and lateral P inputs. Inorganic P inputs from WWTPs within the MLRC subbasin represented less than a third of P load in the Loire at S1 (0.3t P day$^{-1}$ compared to 1 t P day$^{-1}$) despite the presence of 2 10$^6$ inhabitants equivalent within the sub-catchment. Particulate inorganic P represented a very small amount of total inorganic P, and most of it was balanced between erosion and sedimentation processes. The river bed acted like a source

of inorganic P (299 kg P day$^{-1}$).

iii)      Dissolved silica fluxes were slightly affected by phytoplankton activity: 20% of the flux at S1 was assimilated by diatoms. Lateral streams contribution represented 13% of the flux quantified at S2. Phytoplankton increased 4-fold between S1 and S2 during summer blooms, from 4.3 to 17.1 kg C day$^{-1}$. However this calculation only took into account the surviving cells when the water body reached S2. A larger proportion of phytoplankton grew

but part of it decayed and was eventually recycled: the model estimated that 50% green algae and 25% of diatoms that grew between S1 and S2 decayed. Additionally, approximately 25% diatoms were deposited on the river bed. Since the lateral contributions by the Loire river tributaries were not significant (only 0.1 kg C day$^{-1}$), we can estimate that phytoplankton only grew within the MLRC.

iv)      Dissolved oxygen budget was balanced between S1 and S2 (respectively 192 and 208 t O$_2$ day$^{-1}$), with oxygen

inputs from primary producers (phytoplankton, 136 t O$_2$ day$^{-1}$) similar to oxygen depletion by bacteria and zooplankton respiration processes (137 t O$_2$ day$^{-1}$).

## 5 Discussion

Inter-annual, annual and seasonal variations of the main water quality variables simulated by QUAL-NET corresponded to the observations, proving the efficiency of the model at both transferring the different biogeochemical species and also modelling

the main biogeochemical processes instream when they start to control the river biogeochemical variations. At finer resolutions, QUAL-NET provided reasonable daily variations and was able to estimate biogeochemical variations during short-term and highly impacting events such as storm events occurring in summer during a phytoplankton bloom. These performances were considered good enough to allow us investigate confidently the different processes occurring in the river and discuss the controlling variables of eutrophication in the Loire River. This is highlighted in paragraphs 5.1 and 5.2.

Additionally, QUAL-NET was subject to several weaknesses, and potential improvements could be brought; this is detailed in sections 5.3 and 5.4.

### 5.1 Drivers of eutrophication in the Middle Loire River Corridor

### 5.1.1 Biological versus hydrological control of the river biogeochemistry

The model showed that the Loire River biogeochemistry is the result of complex interactions between nutrients availability

and hydrological variations. In winter, the MLRC was mainly controlled by hydrological processes, and nutrients were simply



transferred downstream, with no noticeable control of biological processes. During lower flow period and water temperature increased, C, N, P and oxygen dynamics were dominated by biological processes. The stream algae were clearly P-limited and never reached N or Si limitations, supporting previous studies (Descy et al., 2011; Minaudo et al., 2015). In the MLRC, lateral inputs during summer were not significant compared to the magnitude of fluxes within the Loire River main stem and the

5 intensity of the processes that occurred. The highest phytoplankton concentration was not necessarily observed at the Corridor outlet: during a phytoplankton bloom, P was often depleted before the water could reach S2, and when this occurred, lower phytoplankton growth and mortality rates started to cause a decline of phytoplankton concentration. This maintained low SRP concentrations downstream the point where phytoplankton started to be P-limited, and bacterial activity caused the decrease of oxygen concentration (Li et al., 2014). When a storm event entered the Middle Loire system, the phytoplankton colony

developed in the lower part of the Corridor was flushed downstream, and, as long as physical conditions for phytoplankton growth remained degraded (shorter transit time, increased turbidity), available P was not totally assimilated by phytoplankton, and the river discharged higher SRP concentration downstream S2 in addition to a peak of suspended solids, a vector for particulate P that might be partly available for the biomass further downstream due to desorption processes.

### 5.1.2 P recycling within the Middle Loire River Corridor

Most inorganic P entering the MLRC was assimilated by phytoplankton and bacteria biomasses. However, mineralization of organic matter in summer constituted a significant source of inorganic P. The model suggested that P originating from mineralization represented a P share equivalent to all fluxes entering the MLRC (point and non-point sources) ($\approx$ 1.3 tP day$^{-1}$). Additionally, 40% of the phytoplankton that grew between S1 and S2 was lost due to P-limitation. It suggested that a large portion of inorganic P recycled within the water during the transfer through the MLRC (from S1 to S2). In summer, SRP was

most of the time completely assimilated by phytoplankton, but the phytoplankton subjected to mortality could eventually be partially recycled and constitute a new source of available P. Re-mineralization of autochthonous labile organic particulate P, known as part of the 'microbial loop', is described in the literature of phytoplankton ecology (Li et al., 2014; Reynolds, 2006) and mostly identified in lakes, reservoirs or estuarine systems (James and Larson, 2008; Jossette et al., 1999; Song and Burgin, 2017) but also in rivers (Withers and Jarvie, 2008). On the one hand bacteria compete with phytoplankton for SRP availability,

and on the other hand, bacterial mineralization recycles P and supports phytoplankton growth. These observations, sparsely documented in rivers, comfort the necessity of considering bacterial activity as a major driver of carbon cycling in large eutrophic rivers.

### 5.2 High temporal resolution is needed in water quality models

Model QUAL-NET identified several key processes occurring over a fine temporal scale such as diel fluctuations of SRP

(daily variations oscillated between 0 and 15 µgP L-1 during phytoplankton blooms) and of dissolved oxygen. Diel fluctuations of $O_2$ were often observed and described in previous studies, directly linked to primary producers' activity (Moatar et al., 2001;

Rode et al., 2016; Wade et al., 2012). Sub-daily fluctuations of inorganic phosphorus are sparsely observed, but this is due to limited measurements of high-frequency variations of P concentration. Similar diel fluctuations were found in some other lowland eutrophic rivers; but these cycles were mostly explained as a balance between P contributions from direct sources and non-point sources (Wade et al., 2012).

In the case of the Loire River, model QUAL-NET simulates these diel fluctuations due to a complex interaction between biological uptake and P inputs from instream mineralization, lateral and point-sources inputs or diffusion from the benthos: phytoplankton growth rates during the night is nil, while lateral contributions (both point and non-point sources) still occur, and P keeps being diffused from the benthic compartment, resulting in an increased SRP concentration in the water column. After sunrise, as soon as the biological compartment starts to assimilate more P than the amount of P originating from the

different P sources, SRP concentration starts to decrease again. These subtle variations, revealed by the model, could not be seen based on the daily-scale survey and need to be confirmed with higher-frequency sampling measurements.

High-frequency measurements were also needed to validate the complex interactions between hydrological variations and P-availability simulated by the model when a storm event occurs during a period of phytoplankton bloom.

### 5.3 Sensitivity to phosphorus sorption/desorption representation

During the calibration step, QUAL-NET showed a high sensitivity to the formulation of phosphorus sorption/desorption processes. Compared to other studies using the same formulation, the optimized values found for our study appeared relatively close to the values determined experimentally in the Seine River (Table 1). However, the large variability in the results when one of these two coefficients was modified questioned the use of the model with the current values: if modifications are conducted on the model (in terms of data inputs and/or processes), these coefficients must be re-calibrated. This appears to be

an important weakness in the model until an experimental survey is deployed to assess the spatial and temporal variations in the Loire River of P sorption-desorption characteristics according to Langmuir equilibrium concept.

### 5.4 Issues and potential improvements for model QUAL-NET

Results showed that the deterministic approach provided many useful insights to understand the biogeochemical functioning of the river and the interaction between hydrological and biological factors that control the river biogeochemistry. Some

25 improvements could be made on the model, and the following paragraph lists what appeared to us as the most important changes that could be made.



### 5.4.1 Conflicting time steps between forcing variables and output resolution

The use of high temporal resolution in QUAL-NET proved its usefulness to model processes that occur on a fine temporal scale. However, the only forcing variables with such a fine resolution were the meteorological variables, allowing to compute hourly water temperature and light availability in the water column.

Flows, and therefore water depth and velocity, were daily based, and spatial discretization for discharge was based on catchments that were on average 27 km². Flows within each of these 17 catchments were redistributed into the hydrographic network according to the corresponding drainage area of each river reach. If this method might provide reasonable values of average flow in each stream of the river network, it considers the temporal dynamic to be simultaneous within each of the 17 catchments. This, misses a certain spatial heterogeneity in terms of flow contribution and sources, and also might provoke

conflicting propagation of the signal from headwaters to downstream during storm events and high-flow periods. A semi-distributed hydrological model could address some potential propagation issues during storm events, even if the output frequency remains daily because of the lack of observations on a sub-daily basis.

Nutrients fluxes discharged from point sources were considered constant through time. Waste water treatment plants efficiency in treating sewage can be seasonal (biological processes, variation of population in touristy areas…) and sometimes highly

impacted by storm events. Therefore, we urge local and national water basins authorities to provide at least monthly concentrations and fluxes for the different waste water treatment plants, especially for plants treating sewage from the biggest cities.

Non-point sources concentrations were constant through time. Therefore, it was hypothesized that only hydrological variability drove its input. This representation proved its reliability with a 10-days time step (Garnier et al., 2002), but misses many

processes occurring at least at the seasonal scale, such as for instance nutrient retention by the riparian vegetation during spring and summer (Peterjohn and Correll, 1984; Pinay et al., 1993), denitrification increased during warmer conditions, peaks of nutrient concentrations during soil-rewetting events and when groundwater connects with streams (Dupas et al., 2015a, 2015b). QUAL-NET proved to be efficient to model in-stream processes and would certainly benefit if coupled with land-use models that predict more reliably nutrient non-point inputs such as SWAT (Douglas-Mankin et al., 2010), or HSPF (Fonseca et al.,

2014). This would allow to model the biogeochemical variations for the whole drainage system, not forcing the system with daily-scale measurements at S1, but instead, modelling water quality in the entire basin at S2. To upscale the model to the entire Loire Basin, the influence of lakes and reservoirs have to be considered since they largely modify the transfer of nutrients downstream. This raises another issue, because the connection between streams/rivers with lakes/reservoirs is hardly considered in water quality models at the catchment scale.



### 5.4.1 New eco-hydrological issues that should be considered

In eutrophic rivers, several recent studies clearly showed the increasing concern on the Asian *Corbicula* clams *spp.* that invaded the river networks in South and North America and later in Europe over the past decades (Cataldo and Boltovskoy, 1998; Cohen et al., 1984; Phelps, 1994; Pigneur et al., 2014). This clam plays a significant role in the dynamic of phytoplankton (and

thus, on nutrients) for several rivers in Europe, and for instance seems to be responsible for 70% decrease in the phytoplankton biomass of the Meuse River (Pigneur et al., 2014). The main issue to take into account this grazer is the lack of dataset, both spatially and temporally. In the Loire River, Descy et al. (2011) determined that a *Corbicula* population density of 2.5 to 10 g C $m^{-2}$ was needed to explain the phytoplankton variations, but clams density was then uniformly distributed depending on the river reach due to lack of data. This was not tested in QUAL-NET yet, since very few surveys have been conducted, and spatial

distributions of *Corbicula* spp. population are still unknown.

In addition, aquatic fixed vegetation are able to extract nutrients from the sediment and might keep growing even if the phytoplankton has reached its phosphorus limitation in the water column (Carignan and Kalff, 1980; Hood, 2012). Thus, despite low P availability, macrophytes might keep growing, especially when there is a high P legacy in the river bed sediments. We lack data about macrophytes in the Loire River, but a few unpublished observations in the MLRC presented very significant

densities of *Ranunculus fluitans*, *Myriophyllum spicatum* and *Elodea nuttallii* (Michel Chantereau, personal comm.). Their impact on the Loire River biogeochemistry could be significant, and further developments in the model QUAL-NET should be able to model this biological compartment. However, a reliable monitoring has to be set up, at least in the MLRC.

### 6 Conclusions

The deterministic modelling approach we developed allowed to disentangle the interactions existing between hydrological

processes and biological activity in the Loire River. Results from model QUAL-NET fitted the available daily observations, and the main driving processes could be identified. The Middle Loire River Corridor functions as a biogeochemical reactor in summer during low water period. The system clearly reaches a P-limitation, and our model indicate that internal loadings of P due to bacterial mineralization enhance phytoplankton blooms. The use of high temporal resolution allowed to study the impact of a storm event during a phytoplankton bloom, and identified large diel fluctuations for C, P and $O_2$, but these variations still

need to be confronted to high-frequency in situ measurements. QUAL-NET simulated realistic sub-daily variations from low-frequency forcing variables, and could be applied at a larger scale (e.g. the entire Loire Basin, 110 $10^3$ km²). It could be used to study past evolutions using low frequency dataset as data input, or predict future evolutions under climate change and land use scenarios.





**Author contribution**

C. Minaudo and F. Curie designed the model structure. C. Minaudo and Y. Jullian developed the model code. F. Moatar and N. Gassama and C. Minaudo designed and conducted the daily sampling and the chemical analysis. C. Minaudo prepared the manuscript with contributions from all co-authors.

5 **Acknowledgements**

This work started within the "Eutrophisation-Trends" project (funds from "Agence de l'Eau Loire Bretagne, "Plan Loire Grandeur Nature" and FEDER European funds) and continued within the project "Risque Eutrophisation Plans d'Eau" funded by "l'Office Nationale des Eaux et Milieux Aquatiques". Authors are very thankful to Gilles Billen (Paris Sorbonne-UPMC Univ.) for sharing with us the code of model RIVE and for his careful read of our manuscript. Comments and suggestions from
10 Gilles Pinay (Rennes 1 Univ.) also contributed to improve the manuscript.




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



**Table 1. Values optimized for TSS and Langmuir coefficients during the calibration step, and compared to other values found in other studies**

| | RMSE | Coefficient name | Unit | Optimized value for this study | Values found in other studies for rivers or streams |
|---|---|---|---|---|---|
| TSS | 15 mg L$^{-1}$ | $Veli_0$ | mg TSS L$^{-1}$ | 20 | 20[1] |
| | | $Veli_1$ | mg TSS L$^{-1}$ | 500 | 50[1] |
| | | $Vs_{TSS}$ | m h$^{-1}$ | 0.1 | 0.5[1] |
| SRP | 14 µg L$^{-1}$ | $Kpads$ | mg P L$^{-1}$ | 0.15 | 0.68[1] 0.04[2] 0.01[3] 1.89 to 200[4] |
| | | $Pac$ | mg P (g TSS)$^{-1}$ | 5.5 | 5.6[1] 3.1[2] 12.8[3] 0.3 to 3.0[4] |

1. Billen et al., (1994), Seine River, France
2. Aissa-Grouz, (2015), Seine River, France
3. Vilmin et al., (2015), Seine River, France
4. Jalali and Peikam, (2013), Abshineh River, Iran



**Table 2. Model performances (bias ± s.d. errors) for different time scales: over the entire period of validation (August 1st 2012 to July 31st 2014), in "summer" (April to October) and "winter" (November to March).**

| element | unit | entire period | "summer" | "winter" |
|---|---|---|---|---|
| TSS | mg L-1 | 7.6 ± 13 | 5.4 ± 11 | 10.3 ± 14.8 |
| $NO_3^-$ | mg N L-1 | 0.1 ± 0.4 | 0.1 ± 0.4 | 0.1 ± 0.5 |
| SRP | μg P L-1 | -2 ± 14 | -2.2 ± 15 | -1.9 ± 13 |
| Si | mg Si L-1 | 0.2 ± 1.7 | 0.4 ± 1.3 | -0.1 ± 2.1 |
| PHY | mg C L-1 | 0.0 ± 0.4 | -0.1 ± 0.5 | 0.1 ± 0.1 |
| POC | mg C L-1 | 0.3 ± 1.0 | 0.0 ± 1.1 | 0.6 ± 0.7 |
| DOC | mg C L-1 | 0.4 ± 1.5 | 0.2 ± 1.3 | 0.7 ± 1.8 |



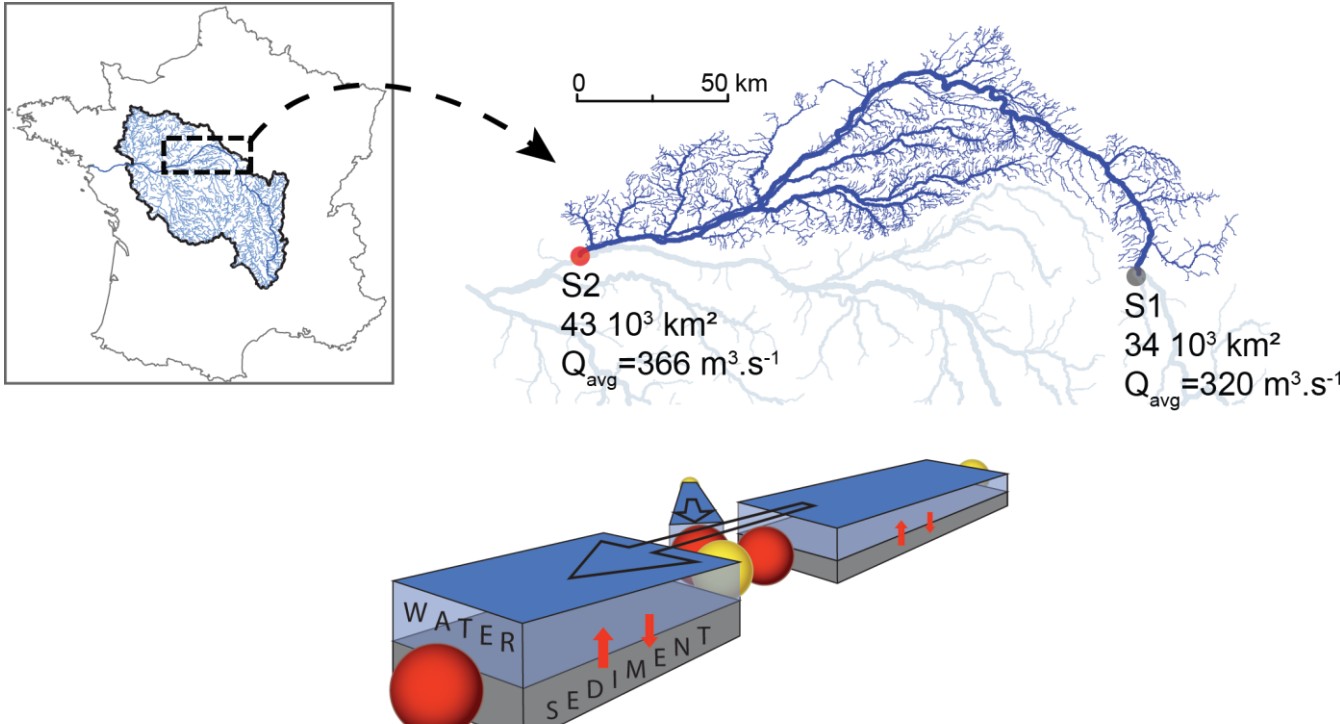

**Figure 1. Study area, i.e. the Middle Loire Corridor sub-catchment defined between stations S1 and S2, and network topology concept used in the model.**



**Figure 2. Architecture of QUAL-NET and sources for the different types of forcing variables.**





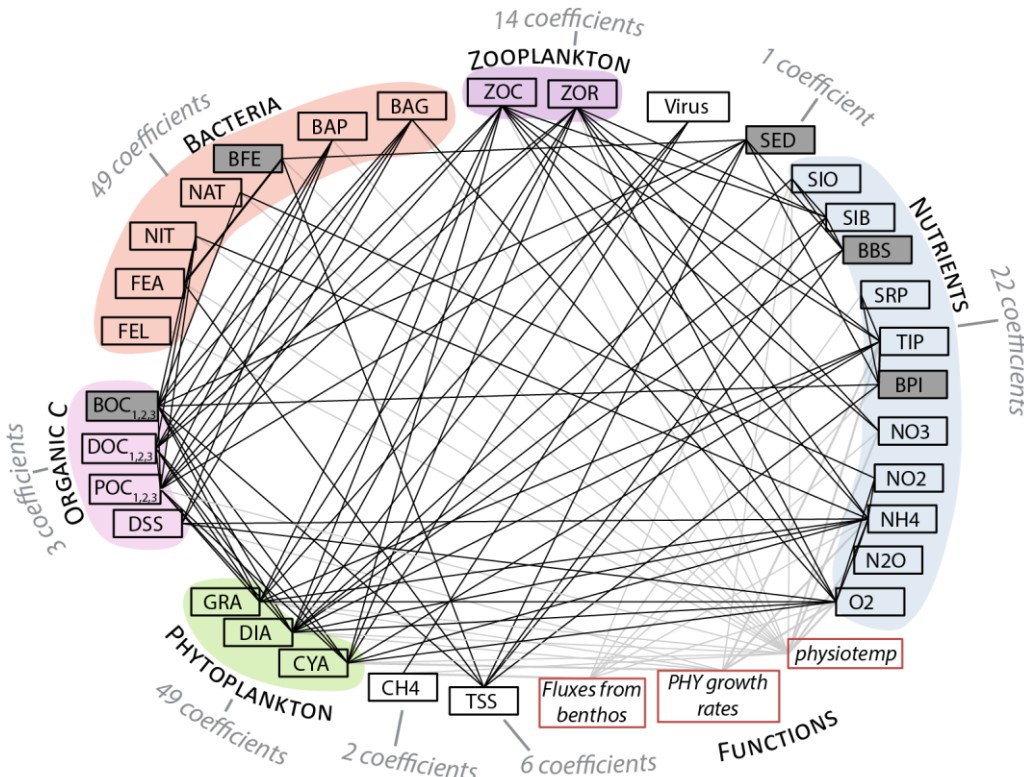

**Figure 3. Main variables interdependency in the biogeochemical model RIVE and associated counted coefficients. Grey plain rectangles identify variables in the benthos component, red rectangles are generic functions often called within the code.**





**Figure 4. Results at station S2 after calibration for the main variables in the model: phytoplankton (PHY), nitrate (NO$_3^-$), dissolved silica (Si), soluble reactive phosphorus (SRP), total suspended solids (TSS) and discharge (Q), particulate organic carbon (POC), dissolved organic carbon (DOC), dissolved oxygen (O$_2$). Last row zooms in on July 2013 for SRP and O$_2$ concentrations to show diel**

5 **fluctuations.**



**Figure 5. Lagrangian view from S1 to S2 of TSS, NO₃⁻, SRP, Si, PHY and O₂ during two selected events. For the summer event are also displayed on the right axis P input from mineralization processes, P uptake from phytoplankton, phytoplankton growth, sedimentation and mortality. For the winter event, the model estimated that water left S1 on February 9th 2013 at midnight and reached S2 on February 10th at 1pm. For the summer event, water left S1 on July 10th 2012 at midnight and reached S2 on July 13th at 1am.**

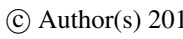



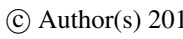

**Figure 6.** Longitudinal evolution of discharge Q, TSS, SRP and PHY concentrations when a storm event occurred between August 8th and August 19th 2013.



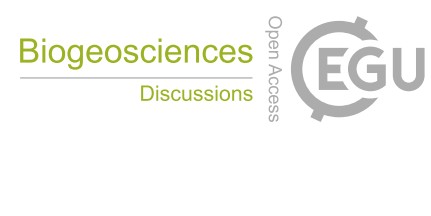

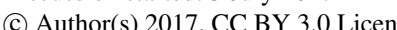

**Figure 7. Average "winter" and "summer" budgets between S1 and S2 for TSS, nitrate, inorganic P, dissolved silica, phytoplankton biomass and dissolved oxygen. Arrow widths are all proportional to calculated fluxes, allowing the visual comparison between "winter" and "summer" periods.**