# Peer review of "QUAL-NET, a high temporal resolution eutrophication model in large hydrographic networks"

_Biogeosciences, 2017_

## Referee Comment (RC1) · Anonymous Referee #1 · 14 Aug 2017

This manuscript details a new modeling framework that was implemented in the Loire River catchment to simulate biogeochemical transformations and pathways in the water column and river sediments. The model incorporates previously developed systems into a more high temporal resolution framework to look at short-term variations of nutrient dynamics and phytoplankton growth. The results presented show that like previous studies, phosphorus availability dictates phytoplankton growth, and that remineralization of organic phosphorus is an important pathway to support phytoplankton growth in the summer time. In addition, the authors propose that physical transport dictates the distribution of nutrients and TSS in the winter time during high flows, and that in lower flow periods temperature dependent biological reactions become more important. The results presented are interesting and in my opinion worthy of publication, and the mod-

eling system can be useful for future work. There are some substantial omissions and corrections, however, that need to be addressed before the manuscript is accepted for publication. To be forthright, I am not a stream ecologist but do have general knowledge of stream ecology. I am relatively versed in hydrodynamic and biogeochemical modeling and therefore will comment mostly on the quality of the modeling work presented.

In general, the author's propose that downstream physical transport dictates the relatively uniform distribution of model constituents in the winter and that biological processes are much more important in the summer. The transit times are only different by about 1.5 days. In my opinion the difference in time is very small in relation to the time it takes for biological reactions to occur such as organic matter remineralization. Without the authors presenting any reaction rates, especially for P remineralization and uptake, it is hard to judge whether the author's physics vs biology explanation is supported. In addition, the authors neglect to discuss temperature variation in the two seasons and the impact it would have on the reaction rates and phytoplankton growth. I would argue that low temperature and thus reactivity of constituents in the winter months can largely determine the distribution in the reach of the river. A thorough explanation of reaction rates and there relation to transport rates would help address these issues. In addition, the authors could do a model run with constant flow and see if the same patterns emerge. If they do, then their conclusion would be supported.

Another result that needs some more explanation is an important source of phosphorus in summer coming from remineralization. What is the source of the organic matter that is fueling the enhanced release of phosphorus? Is it autochthonous to this river reach, from sediments, from the watershed? More detail on the source of the P needs to be added, because if the reach of the river was in a steady state during summer i.e. recycling from algae, there wouldn't necessarily be algae blooms; the population would be constant in time. The budget diagrams are extremely useful and seem to be underutilized in the manuscript. Highlighting the P budget and the different processes

further in the results section would hopefully aid in clearing up some of the ambiguity associated with the budget.

Lastly, how the sediment and water column interact biologically and chemically needs to be further explained. What is the sediment model and how is it coupled with the water column? What happens to porewater in the stream sediment when it is resuspended during erosion?

In general, the writing style needs to be edited for consistency between sections and grammar. The grammar is at times unclear and incorrect, and therefore needs to be addressed. I recommend having a native English speaker edit this manuscript as there are many small grammatical errors.

Line Comments:

Abstract

Page 1, line 8: sentence is awkward, reword

1,11: insert "that is" before "prone"

1,18: change "or" to "and"

Introduction

1,24: second part of sentence doesn't make sense

2,27: not all streams will be drier, rainfall may increase in many parts of the world

2,29-30: this is a good point

Study Site

3,13: change "respectively on Figure 1" to "(Fig. 1)"

3,17: remove "some"

3,18: change "were since divided 2.5-fold" to "decreased by 2.5 times"

3,19: "time to time" how often

Methods

4,2: change "dynamic" to "dynamics"

4,3: remove "the" before "phytoplankton"

4,11: Confusing sentence, please revise

4,11-13: How long does a model simulation take and on what platform? More information could be helpful to the reader to see if this is a tool they might want to use in the future

4,16: In figure 2, it seems to me that the deltaT and deltaX are backwards

4,17: what do authors mean "nutrient sources forcing variables"? revise

4,15-25: how do the nutrients get added into the model domain, specifically?

6,15: remove "and" before "analyzer"

6,25: upper vs lower nodes; it confuses me which one is which. I would think "upper" would be connected to one lower node, and "lower" nodes would be connected to two upper. Maybe this is a terminology issue?

6,28: sediment is mentioned again but the manuscript lacks any details about the sediment model. A section that details the sediment model and describes how it is coupled to the water column seems important.

6,29: the sediment initial condition being homogenous is likely ok, but without information about the model itself it is hard to assess the appropriateness of this initialization. Were any tests done to show this was appropriate?

7,5: This sentence is confusing, what doesn't get transferred downstream? Sediment?

7,14-15: Why were the WWTP locations not known? Surely the coordinates exist?

7,29-30: Was this optimized numerically or by hand (manually)?

Results

9,5: how was the Lagrangian view captured, specifically? How was the water mass tracked?

9,20: change "on" to "in"

9,27: First mention of statistics, how do you calculated bias and error?

10,2: Does the lack of the ability for the model to capture storms complicate the interpretation of the storm flow results in section 4.4?

Page 10: This entire section reiterates information in Table 2, it can probably be summarized in a sentence or two.

10,21: It is interested that DOC varied with flow, and flow is seasonal, but the DOC concentration wasn't seasonal. Maybe expand on this a little bit more...

11,6: In figure 5, it is curious to me that the phytoplankton are growing at night. Shouldn't primary production go to 0, or is this a different measure of growth?

Discussion

14,9-14:

14,18: "lost due to P-limitation" what do the authors mean by lost? Clarify

15,15-21: Can the authors quantify how sensitive the model was to these parameters? Can the authors speculate how useful this parameterization would be? Similar river systems, similar environments or would the model always have to be recalibrated?

---

## Referee Comment (RC2) · Anonymous Referee #2 · 27 Nov 2017

General comments:

This paper concentrates on identifying important temporal drivers for drainage networks water quality with a fine temporal resolution at the regional scale. The authors try to explain that the so called QAL-NET model could compute the biogeochemical processes and simulate the eutrophication event in the Middle Loire River Corridor. They conclude that phytoplankton variations in the Loire River were governed by phosphorus availability and transit time. Their modeling study found that continuous phytoplankton blooms occurred in the study area were triggered by the recycled of phytoplankton cells growing in the upper part of the studied corridor through the microbial loop. While this result is not surprising finding, the approach and method used may be valuable to be published. Two general suggestions are highlighted below:

[Figure]

1. The hypothesis and purpose of the study is somehow unclear. I do not really understand what the objective of this paper. Does the paper focus on the new modeling approach or the eutrophication in the modelling study?

2. I found the manuscript written with unclear messages. The manuscript seems were written without final editing. I think it needs a language editing. Also, please avoid repetition of adverb such as "yet" and "additionally" in the text.

Specific comments:

1. The manuscript states that most of biogeochemical processes are water temperature dependent, however, I found that it does not provide modeling result on temperature variable. How does the daily temperature look like? During the travel time from S1 to S2, does it highly fluctuated? During summer, does the temperature at S2 close to the temperature value at S1?

2. The fluxes and concentration of point sources were considered constant over the time in the model. Further explanation on how much and how fluxes and concentration were estimated is needed.

3. The manuscript does not discuss how the model treats the nutrient source coming from resuspeded sediment and nutrient fluxes between water and sediment interface. I think A paragraph discussing this would be helpful for the reader.

Technical comments:

(Page 1: Line 19) Change "or" to "end"

(2:15-30) "Yet" and "additionally" adverbs were used extensively.

(2:31) Instead of "context", perhaps use "study"?

(3:3, 7, 26) Missing multiply mark "x". Also, in the figure 1.

(3: 18-20) Please reorganize these unclear sentences.

(3: 22) Change "the fusion" to "a couple"

(4: 16) In Figure 2, switch delta x with delta t.

(7: 19) In Figure 3, change the color lines and add a list of abbreviations to improve the figure clarity.

(7: 31) What and how many variables were calibrated?

(page 11, 12, and 13) I do not think lower roman numbering is necessary in the text.

(13, 21) Consider improving "At finer resolution" words in the conclusion. What resolution? Time or space? Finer from what?

---

## Author Comment (AC2) · 25 Jan 2018

Authors are grateful for comments and suggestions from Referee 2. All raised issues were listed below and carefully answered.

//—- R2C1: The hypothesis and purpose of the study is somehow unclear. I do not really understand what the objective of this paper. Does the paper focus on the new modeling approach or the eutrophication in the modelling study?

//—- A: The main objective was to assess the hydrological versus biological control of water quality in a eutrophic system. We proposed an original model to determine the controlling factors based on high temporal frequency. Thus, presenting the new model approach had to be a second objective in this paper.

[Figure]

//—- R2C2: I found the manuscript written with unclear messages. The manuscript seems were written without final editing. I think it needs a language editing. Also, please avoid repetition of adverb such as "yet" and "additionally" in the text.

//—- A: A native speaker went carefully through the manuscript to clarify as much as possible our messages.

//—- R2C3: The manuscript states that most of biogeochemical processes are water temperature dependent, however, I found that it does not provide modeling result on temperature variable. How does the daily temperature look like? During the travel time from S1 to S2, does it highly fluctuated? During summer, does the temperature at S2 close to the temperature value at S1?

//—- A: That is correct. We agree that presenting results of water temperature estimations is necessary. Water temperature was highly seasonal and fluctuates between 0 and 30°C (Figure A8). In summer in the Loire River, amplitude of diel cycles ranged between 0.2 and 1.5°C. Seasonal variations between S1 and S2 were very close (Figure A9). Temperature variations at the daily scale were highly contrasted at the two stations, highlighting meteorological and hydrological controls on water temperature.

//—- R2C4: The fluxes and concentration of point sources were considered constant over the time in the model. Further explanation on how much and how fluxes and concentration were estimated is needed.

//—- A: The regional Water Agency ("Agence de l'Eau Loire Bretagne") publishes N-P-C and total effluent fluxes exiting WWTP for all domestic and industrial effluents. It was estimated in 2010 that point sources represent in the Middle Loire sub-catchment (our study) 322 kgP day-1 and 1.9 tN day-1. Model QUAL-NET uses directly this data. We would add this information to the manuscript.

//—- R2C5: The manuscript does not discuss how the model treats the nutrient source coming from re-suspended sediment and nutrient fluxes between water and sediment

interface. I think a paragraph discussing this would be helpful for the reader.

//—- A: This might have been unclear in our manuscript. The model estimates for each river reach and at each time-step quantities of suspended particles eroded or that settled on the river bed (based on sedimentation velocities). Particles are both inorganic and organic with three levels of lability. Re-suspension might fuel the water column with soluble reactive phosphorus via desorption processes from suspended matter.

Diffusion processes for nutrients between the two layers are also considered. The benthic compartment can be either a source or a sink of nutrients, depending on redox conditions. All these processes are modeled using Billen et al. 2014 (Ann. Limnol). Equations in this formulation provided estimates of NH4, NO3, PO4, SiO2 and O2 fluxes across the water – sediment interface. The sediment layer was split into two sub-layers. The one at the bottom was considered compact and not erodible, the other one could potentially be re-suspended. Nutrient fluxes between these two sediment layers were also considered in our model.

//—- R2C6: (Page 1: Line 19) Change "or" to "end" //—- A: there was no "or" page 1 line 19

//—- R2C7: (2:15-30) "Yet" and "additionally" adverbs were used extensively. //—- A: We carefully read the manuscript and tried to avoid these adverbs.

//—- R2C8: (2:31) Instead of "context", perhaps use "study"? //—- A: We really meant "context".

//—- R2C9: (3:3, 7, 26) Missing multiply mark "x". Also, in the figure 1. //—- A: OK, this was modified.

//—- R2C10: (3: 18-20) Please reorganize these unclear sentences. //—- A: OK "Since 1990, phosphorus concentrations were divided 2.5-fold and phytoplankton blooms declined 3-fold (Floury et al., 2012; Minaudo et al., 2015; Oudin et al., 2009). Algal

blooms are still occurring from time to time, questioning the source of phosphorus."

//—- R2C11: (3: 22) Change "the fusion" to "a couple" //—- A: We changed it to: "It is the coupling between a thermal model T-NET (Beaufort et al., 2016), and a biogeo-chemical model, RIVE (Garnier et al., 2002)."

//—- R2C12: (4: 16) In Figure 2, switch delta x with delta t. //—- A: OK

//—- R2C13: (7: 19) In Figure 3, change the color lines and add a list of abbreviations to improve the figure clarity. //—- A: We don't think this is needed: variable names already have abbreviations and are organized. Our objective with this figure really was to show the complexity of our model and the fact that variables are highly inter-dependent.

//—- R2C14: (7: 31) What and how many variables were calibrated? //—- A: Two variables were calibrated: TSS and Total Inorganic P concentrations. To achieve this, 5 coefficients in total were manually calibrated (see Table 1 in the manuscript).

//—- R2C15: (page 11, 12, and 13) I do not think lower roman numbering is necessary in the text. //—- A: OK

//—- R2C16: (13, 21) Consider improving "At finer resolution" words in the conclusion. What resolution? Time or space? Finer from what? //—- A: You are right, we meant "higher temporal resolution"

// ————————————- FIGURE CAPTIONS ——————————————-//

Figure A8. Water temperature estimated with T-NET module: top panel presents hourly variations at station S2 over the period considered. Bottom panel plots the evolution of water temperature when the water moves downstream from S1 to S2 during summer

Figure A9. Water temperature estimated by T-NET module at S1 and S2

**Fig. 1.** Figure A8. Water temperature estimated with T-NET module. Temporal variations at S2 (top) and Lagrangian view from S1 to S2 in summer

**Fig. 2.** Figure A9. Water temperature estimated by T-NET module at S1 and S2

---

## Author Response (AR1)

**Response to Referees**

Authors are grateful for comments and suggestions from the two referees. All raised issues were listed below and carefully answered.

We had to run several other simulations to address some comments (especially the first comment from Referee 1). This did not affected results in the manuscript. Only descriptions and interpretations were modified.

Referees' comments are shown in blue. Authors' responses are in red.

**Response to referee 1**

Referee 1 Comment 1: In my opinion the difference in time is very small in relation to the time it takes for biological reactions to occur such as organic matter remineralization. Without the authors presenting any reaction rates, especially for P remineralization and uptake, it is hard to judge whether the author's physics vs biology explanation is supported.
In addition, the authors neglect to discuss temperature variation in the two seasons and the impact it would have on the reaction rates and phytoplankton growth. I would argue that low temperature and thus reactivity of constituents in the winter months can largely determine the distribution in the reach of the river. A thorough explanation of reaction rates and there relation to transport rates would help address these issues.
In addition, the authors could do a model run with constant flow and see if the same patterns emerge. If they do, then their conclusion would be supported.
A: We thank Referee 1 for this thoughtful comment. We agree that the difference between travel times in winter compared to summer is too small to fully explain seasonal variations observed at the downstream station S2. We also know that changes in reactivity rates are triggered by warmer water temperature, and this must play a role. Almost every single variable in the model is temperature dependent. Phytoplankton dynamic also depend on light availability (conditioned indirectly by suspended sediment concentrations, governed by hydrological variations) and, of course, nutrients availability.
As suggested by Referee 1, we ran a constant flow simulation with Q in the Loire itself = 200 m3 s-1, and Q = 0.1 m3 s-1 everywhere else. We compared this run with another simulation where Q in the Loire River was 1000 m3 s-1 (Figure A1).

[Figure]

*Figure A1. Phytoplankton concentration at S2 observed and estimated by QUALNET for three different simulations: the reference simulation used in the paper, a constant flow run with Q = 200 m3s-1 in the Loire River, and a constant flow run with Q = 1000 m3s-1 in the Loire River.*

Results showed for all simulations strong seasonal variations with phytoplankton blooms in summer and very low phytoplankton concentration in winter. Phytoplankton development was similar between the reference simulation and the constant low-flow simulation. However, results with a constant high-flow presented much lower PHY concentrations. This proved how much travel time impacts phytoplankton blooms occurrences. However, the fact that phytoplankton concentration remained low during winter with constant low-flow conditions proves that Q can't be the only key driver, especially because nutrient concentrations are highest in winter.

We also ran a simulation with normal flow variations but constant water temperature throughout the entire period with T = 13.7°C (i.e. the median water temperature simulated in the Loire River by T-NET module).

[Figure]

*Figure A2. Phytoplankton concentration at S2 observed and estimated by QUALNET for three different simulations: the reference simulation used in the paper, and a constant temperature run with T = 13.7°C everywhere in the Loire River*

Results showed (Figure A2) that the intensity of PHY (peaks values) was sensitive to water temperature: we observed lower PHY concentrations in the constant T°C run. However, the dynamic of PHY remained very close to the reference simulation, proving that water temperature, just like travel time, can't be qualified as the main driver of PHY variations. Phytoplankton variations in the Loire River are co-controlled by Q, T°C, nutrients and light availabilities, and all these variables interact with each other.

Viewed in a Lagrangian way during a summer event (starting date at Q1 = 17th July, 2012same date as in the paper on Figure 5), we observed that phytoplankton development was much more affected by shorter travel times (run with Q = 1000 m3 s-1) than with colder water temperature (see Figure A3 below). P availability played a major role, and P exhaustion was reached 2.5 days after the starting date from S1 for all simulations except with the high-flow simulation where no P limitation was simulated, because travel time from S1 to S2 was too short.

[Figure]

*Figure A3. Lagrangian view from S1 to S2 of phytoplankton and PO4 concentrations for 4 different scenarios: the reference simulation, a constant T°C simulation where T = 13.7°C in the Loire River, and two constant flow simulations where Q = 200 or 1000 m3 s-1*

It was decided to add Figure A3 as a Supplement file S1.

R1C2: What is the source of the organic matter that is fueling the enhanced release of phosphorus? Is it autochthonous to this river reach, from sediments, from the watershed? More detail on the source of the P needs to be added,

because if the reach of the river was in a steady state during summer i.e. recycling from algae, there wouldn't necessarily be algae blooms; the population would be constant in time.

A: Large amounts of organic matter enter the Middle Loire at its upper limit S1: it is estimated with our daily measurements that approximately 80t of organic C enter the system at S1 every day under low flow periods (see also Minaudo et al. 2016 in Environmental Monitoring Assessment). Approximately 80% of it is dissolved organic carbon, the rest is particulate. Model QUAL-NET tells us that a significant proportion of DOC is bioavailable and consumed by heterotrophic bacteria (16 tC day-1 in summer, see Figure A4). Part of this organic matter is eventually mineralized, depending on oxygen conditions. This constitutes another pathway for P, and, combined with P recycling processes from dead algae, it explains that blooms may still occur despite P limited conditions.

These processes are explicitly represented in the model, and can be seen in the C budget, as depicted in the figure below. This also highlights how important it is to explicitly represent bacteria in our water quality models.

We would add this analysis to our manuscript.

[Figure]

*Figure A4. DOC and POC budgets assessed with QUAL-NET between S1 and S2*

Figure A4 was added to Figure 7.

R1C3: How the sediment and water column interact biologically and chemically needs to be further explained. What is the sediment model and how is it coupled with the water column? What happens to porewater in the stream sediment when it is resuspended during erosion?

A: the following interactions between the sediment layer and the water column are considered:
- Sedimentation/erosion processes of particles depending on flow energy. Particles are both inorganic and organic with three levels of lability.
- Diffusion processes for nutrients between the two layers. The benthic compartment can be either a source or a sink of nutrients, depending on redox conditions. All these processes were modeled using Billen et al. 2014 (Ann. Limnol). Equations in this formulation provided estimates of NH4, NO3, PO4, SiO2 and O2 fluxes across the water – sediment interface. The sediment layer was split into two sub-layers. The one at the bottom is considered compact and not erodible, the other one might potentially be re-suspended. Nutrient fluxes between these two sediment layers were also considered in our model.

The sediment model is a simple power law model based on the flow velocity. Equations are already explained in section 3.3.1, thus we did not believe it had to be clarified.

Although fluxes from and to the benthic compartment were considered (see above), pore-water was not explicitly considered as an object in our model. Only the physical dynamic of sediment particles was considered.

This was modified in the manuscript, refer to page 7 lines 5-11 of the marked-up manuscript.

R1C4: How long does a model simulation take and on what platform? More information could be helpful to the reader to see if this is a tool they might want to use in the future

A: It takes approximately 4 hours to simulate hourly biogeochemical evolutions of 3361 stream segments over a 3 year period on a 2 processors platform (Intel(R) Xeon(R) CPU E5-2670 0 @ 2.60GHz) with 16 cores (64 Go, DDR3 = 1600 MHz). Computing time could be reduced on a more efficient platform.

This was modified in the manuscript, refer to page 7 lines 14-16 of the the marked-up manuscript.

R1C5: Why were the WWTP locations not known? Surely the coordinates exist?
A: Coordinates of WWTP buildings are well known, but not the exact location of WWTP discharge points for all plants in the studied zone. That is why we had to make assumptions.
This was modified in the manuscript, refer to page 11 lines 1-2 of the marked-up manuscript.

R1C6: Page 7 lines 27-30: Was this optimized numerically or by hand (manually)?
A: All calibration steps were conducted manually based on sensitivity analysis.
This was highlighted in the manuscript, refer to page 12 line 3 of the marked-up manuscript.

R1C7: how was the Lagrangian view captured, specifically? How was the water mass tracked?
A: Lagrangian views were produced based on travel time estimated for each reach and at each time-step. The matrix of travel time was estimated based on known discharge and river morphology (estimated for most reaches, except for the Loire River itself were we used measured values from previous studies).
The following figure A5 explains the successive steps we considered to compute Lagrangian profiles:

[Figure]

Figure A5. Successive steps to produce Lagrangian longitudinal profiles

This was added to the manuscript, refer to page 13 lines 16-17 of the marked-up manuscript.

R1C7: Page 9 lines 27: First mention of statistics, how do you calculated bias and error?
A: Bias and std errors are mentioned in section 3.4 but equations were not shown.

$$Err_{std} = std(observation - model)$$
$$Err_{bias} = \sum_{i=1}^{n} \frac{observation(i) - model(i)}{n}$$

This was added to the manuscript, refer to page 13 lines 6-7 of the marked-up manuscript.

R1C8: Does the lack of the ability for the model to capture storms complicate the interpretation of the storm flow results in section 4.4?
A: We chose to describe the results of a storm event that was satisfactorily predicted on a sediment dynamic point of view (see Figure A6 below). We do not think that because the model underestimates sediment variations for several storms impacted our interpretations in section 4.4.

[Figure]

*Figure A6. Discharge and observed and modeled TSS concentration during the selected storm event.*

R1C9: Page 10: This entire section reiterates information in Table 2, it can probably be summarized in a sentence or two.

A: OK. We still think it is necessary to describe temporal variations over the studied period, but we decided to extract key messages from Table 2 as follows:

"QUAL-NET provided reasonable estimations for the main variables (report to Table 2 for bias and standard deviation errors). Seasonal variations were correctly simulated for all variables. At the scale of the storm event, a few events were observed with the daily survey but were not represented by the model, especially for several events that occurred under low flow conditions. A phytoplankton bloom event at the end of summer 2012 was simulated but this did not correspond to our observations. The model provided interesting diel fluctuations in summer for PHY, SRP and $O_2$ (e.g. SRP concentration fluctuated between 0 and 15 µg P $L^{-1}$), but the reliability of these variations could not be verified with our measurements.

Performances appeared similar between seasons (Table 2) with approximately the same range of errors in winter or summer, except for dissolved silica whose simulated concentrations in winter were subject to higher imprecisions (2.1 against 1.3 mgSi $L^{-1}$ in summer) and for PHY with lower absolute errors in winter (a period with very low PHY concentrations)."

This was added to the manuscript, refer to page 15 lines 15-28 of the marked-up manuscript.

R1C10: page 10, line 21: It is interested that DOC varied with flow, and flow is seasonal, but the DOC concentration wasn't seasonal. Maybe expand on this a little bit more…

A: As it is shown in the DOC budget assessed during high flow and low flow periods (see above response to comment R1C2), DOC is only slightly transformed by biogeochemical processes within the Middle Loire River Corridor. Unlike POC, DOC variations at S2 are very close to variations at S1. QUAL-NET cannot fully explain why DOC isn't seasonal at the entrance of the Middle Loire River Corridor. However, we can hypothesize based on QUAL-NET results that DOC variations are largely driven by upstream soil leaching, and metabolic activities within the water column play only a minor role.

R1C11: page 11, line 6: In figure 5, it is curious to me that the phytoplankton are growing at night. Shouldn't primary production go to 0, or is this a different measure of growth?

A: "phytoplankton growth" in Figure 5 represents phytoplankton growth controlled by the availability of intracellular carbon and nutrients, and not photosynthesis activity which, we agree, goes to zero at night. Phytoplankton growth is mostly driven by water temperature and nutrients availability.

We would add this explanation in the manuscript, along with the reference of the model AQUAPHY (Lancelot et al. 1991) which serves as a basis in QUALNET biogeochemical module to describe primary producers dynamic. This formulation is also used in models RIVERSTRHALER or ProSe.

This was modified in the manuscript, refer to the legend of Figure 5.

R1C12: page 14, line 18: "lost due to P-limitation" what do the authors mean by lost? Clarify

A: "lost" was not the right term. We meant that "PHY concentration declined by 40% due to P-limitation".

R1C13: page 15, lines 15-21: Can the authors quantify how sensitive the model was to these parameters? Can the authors speculate how useful this parameterization would be? Similar river systems, similar environments or would the model always have to be recalibrated?

A: During the calibration step, we observed that the sensitivity to phosphorus sorption/desorption coefficients was large. A previous study (in Camille Minaudo's PhD thesis) describes this sensitivity. In the model, PO4 is determined based on Langmuir equilibrium concept which uses TSS and Total Inorganic P concentrations and two coefficients Kpads and Pac that needs to be either calibrated or measured experimentally. Very different values were found in the literature for these coefficients, and largely impacts the estimation of PO4: Figure A7 shows differences in PO4 estimations for three different sets of values for Kpads and Pac extracted from 3 different studies on the Seine River.

[Figure]

Figure A7. Sensitivity of PO4 estimations from total inorganic P (PIT) and suspended solids concentrations (MES) based on the Langmuir equilibrium concept

Pac and Kapds values have never been assessed experimentally in the Loire River sediment. Our manual calibration found values very close to what AIssa Grouz (2015) has found experimentally in the neighboring Seine basin, showing that our parametrization could be used on other systems. However, if no specific measurements were conducted on the river sediment, we highly recommend to calibrate these coefficients within reasonable ranges.

**Response to referee 2**

R2C1: The hypothesis and purpose of the study is somehow unclear. I do not really understand what the objective of this paper. Does the paper focus on the new modeling approach or the eutrophication in the modelling study?

A: The main objective was to assess the hydrological versus biological control of water quality in a eutrophic system. We proposed an original model to determine the controlling factors based on high temporal frequency. Thus, presenting the new model approach had to be a second objective in this paper.

R2C2: I found the manuscript written with unclear messages. The manuscript seems were written without final editing. I think it needs a language editing. Also, please avoid repetition of adverb such as "yet" and "additionally" in the text.

A: A native speaker went carefully through the manuscript to clarify as much as possible our messages.

R2C3: The manuscript states that most of biogeochemical processes are water temperature dependent, however, I found that it does not provide modeling result on temperature variable. How does the daily temperature look like? During the travel time from S1 to S2, does it highly fluctuated? During summer, does the temperature at S2 close to the temperature value at S1?

A: That is correct. We agree that presenting results of water temperature estimations is necessary.

[Figure]

*Figure A8. Water temperature estimated with T-NET module: top panel presents hourly variations at station S2 over the period considered. Bottom panel plots the evolution of water temperature when the water moves downstream from S1 to S2 during summer*

Water temperature was highly seasonal and fluctuates between 0 and 30°C (Figure A8). In summer in the Loire River, amplitude of diel cycles ranged between 0.2 and 1.5°C.

[Figure]

*Figure A9. Water temperature estimated by T-NET module at S1 and S2*

Seasonal variations between S1 and S2 were very close (Figure A9). Temperature variations at the daily scale were highly contrasted at the two stations, highlighting meteorological and hydrological controls on water temperature.

This was added to the manuscript, refer to Figure 5 in the marked-up manuscript.

R2C4: The fluxes and concentration of point sources were considered constant over the time in the model. Further explanation on how much and how fluxes and concentration were estimated is needed.
A: The regional Water Agency ("Agence de l'Eau Loire Bretagne") publishes N-P-C and total effluent fluxes exiting WWTP for all domestic and industrial effluents.
It was estimated in 2010 that point sources represent in the Middle Loire sub-catchment (our study) 322 kgP day-1 and 1.9 tN day-1. Model QUAL-NET uses directly this data.
We would add this information to the manuscript.
This was added to the manuscript, refer to page 9 lines 12-13 of the marked-up manuscript.

R2C5: The manuscript does not discuss how the model treats the nutrient source coming from re-suspended sediment and nutrient fluxes between water and sediment interface. I think a paragraph discussing this would be helpful for the reader.
A: This might have been unclear in our manuscript.
The model estimates for each river reach and at each time-step quantities of suspended particles eroded or that settled on the river bed (based on sedimentation velocities). Particles are both inorganic and organic with three levels of lability. Re-suspension might fuel the water column with soluble reactive phosphorus via desorption processes from suspended matter.
Diffusion processes for nutrients between the two layers are also considered. The benthic compartment can be either a source or a sink of nutrients, depending on redox conditions. All these processes are modeled using Billen et al. 2014 (Ann. Limnol). Equations in this formulation provided estimates of NH4, NO3, PO4, SiO2 and O2 fluxes across the water – sediment interface. The sediment layer was split into two sub-layers. The one at the bottom was considered compact and not erodible, the other one could potentially be re-suspended. Nutrient fluxes between these two sediment layers were also considered in our model.

This was added to the manuscript, refer to page 7 lines 5-11 of the marked-up manuscript.

R2C6: (Page 1: Line 19) Change "or" to "end"
A: there was no "or" page 1 line 19

R2C7: (2:15-30) "Yet" and "additionally" adverbs were used extensively.
A: We carefully read the manuscript and tried to avoid these adverbs.

R2C8: (2:31) Instead of "context", perhaps use "study"?

A: We really meant "context", but this was modified in the manuscript to:

"The objectives of our study were twofold: firstly, develop a model able to simulate hydrological and biogeochemical processes in drainage networks at the regional scale (over 104 km²), with hourly resolution and water temperature explicitly determined to allow potential climate change impact assessment; secondly, disentangle the different processes involved in eutrophication in a large river and identify their main drivers. To achieve this, the model QUALity-NETwork (QUAL-NET) was developed based on the integration of a biogeochemical model, RIVE (Garnier et al., 2002), in a thermal model, T-NET (Beaufort et al., 2016). This new model was tested on a selected portion of the Loire River basin, the Middle Loire River Corridor, draining 43x103 km², where the river main stem (270 km long) is prone to eutrophication in summer (Descy et al., 2011; Lair and Reyes-Marchant, 1997; Minaudo, 2015; Minaudo et al., 2015)."

Refer to page 4 lines 18-25 of the marked-up manuscript.

R2C9: (3:3, 7, 26) Missing multiply mark "x". Also, in the figure 1.

A: OK, this was modified.

R2C10: (3: 18-20) Please reorganize these unclear sentences.

A: OK

"Chlorophyll-a concentration was often over 250 µg L-1 in the 1980s, and many efforts were conducted since 1990 to limit phosphorus point and non-point sources and counteract eutrophication: since 1990, phosphorus concentrations were divided 2.5-fold and phytoplankton blooms declined 3-fold (Floury et al., 2012; Minaudo et al., 2015; Oudin et al., 2009). Even if phytoplankton in the Loire system is now clearly P-limited, algal blooms still occur (Abonyi et al., 2012), questioning the source of phosphorus and suggesting potential recycling processes."

This was modified, refer to page 6 lines 10-15 of the marked-up manuscript.

R2C11: (3: 22) Change "the fusion" to "a couple"

A: We changed it to:

"It is the coupling between a thermal model T-NET (Beaufort et al., 2016), and a biogeochemical model, RIVE (Garnier et al., 2002)."

This was modified, refer to page 6 lines 17-18 of the marked-up manuscript.

R2C12: (4: 16) In Figure 2, switch delta x with delta t.

A: OK

R2C13: (7: 19) In Figure 3, change the color lines and add a list of abbreviations to improve the figure clarity.

A: We don't think this is needed: variable names already have abbreviations and are organized. Our objective with this figure really was to show the complexity of our model and the fact that variables are highly inter-dependent.

R2C14: (7: 31) What and how many variables were calibrated?

A: Two variables were calibrated: TSS and Total Inorganic P concentrations. To achieve this, 5 coefficients in total were manually calibrated (see Table 1 in the manuscript).

This was added to the manuscript, refer to page 12 lines 4-5 of the marked-up manuscript.

R2C15: (page 11, 12, and 13) I do not think lower roman numbering is necessary in the text.

A: OK

This was modified.

R2C16: (13, 21) Consider improving "At finer resolution" words in the conclusion. What resolution? Time or space? Finer from what?

A: You are right, we meant "higher temporal resolution"

This was modified, refer to page 18 lines 21-22 of the marked-up manuscript.

[revised manuscript text omitted]

0 50 km

43 10³ km²
Q_avg=366 m³ s⁻¹

34 10³ km²
Q_avg=320 m³.s⁻¹

WATER

SEDIMENT

0 50 km

43x10³ km²
Q_avg=366 m³.s⁻¹

34x10³ km²
Q_avg=320 m³.s⁻¹

upper node

lower node

WATER

SEDIMENT

**Figure 1. Study area,  the Middle Loire Corridor sub-catchment defined between stations S1 and S2, and network topology concept used in the model.**

[Figure]

[Figure]

[Figure]

**Figure 2. Architecture of QUAL-NET and  sources .**

[Figure]

**Figure 3.** Main variables interdependency in the biogeochemical model RIVE and associated  coefficients. Grey plain rectangles identify variables describing the benthos component, red rectangles are generic functions often called within the code.

[Figure]

[Figure]

**Figure 4. Results at station S2 after calibration for the main variables in the model: discharge (Q), phytoplankton (PHY), nitrate (NO3⁻), dissolved silica (Si), soluble reactive phosphorus (SRP), water temperature, total suspended solids (TSS) and discharge (Q), particulate organic carbon (POC), dissolved organic carbon (DOC), dissolved oxygen (O₂). Last row zooms in on July 2013 for SRP and O₂ concentrations in July 2013 to show simulated diel fluctuations.**

[Figure]

**Figure 5.** Lagrangian profiles from S1 to S2 of TSS, NO₃⁻, SRP, Si, PHY and O₂ during two selected events. For the summer event, are also displayed on the right axis P input from mineralization processes, P uptake from phytoplankton, phytoplankton growth

5  rate (availability of intracellular carbon and nutrients), sedimentation and mortality rates.

[Figure]

[Figure]

**Figure 6. Longitudinal evolution of discharge Q, TSS, SRP and PHY concentrations when a storm event occurred between August 8th and August 19th 2013.**

[Figure]

| | Winter high flows | Summer low flows |
|---|---|---|
| **TSS** (t day⁻¹) | | |
| **Nitrate** (t N day⁻¹) | | |
| **Inorganic P** (kg P day⁻¹) | | |
| **Si** (t day⁻¹) | | |
| **Phytoplankton** (kg C day⁻¹) | | |
| **O₂** (t day⁻¹) | | |

[Figure]

**Figure 7.** Average "winter" and "summer" budgets between S1 and S2 for TSS, nitrate, inorganic P, dissolved silica, . All arrow widths are  proportional to calculated fluxes, allowing the visual comparison between "winter" and "summer" periods.

[Figure]

**Figure 8. Average "winter" and "summer" budgets between S1 and S2 for phytoplankton biomass, dissolved and particulate organic carbon, and dissolved oxygen. All arrows widths are proportional to calculated fluxes, allowing the visual comparison between "winter" and "summer" periods.**

[Figure]

10/07/2012 0am

Figure S1. Lagrangian profiles from S1 to S2 of phytoplankton and SRP concentrations for four contrasted simulations to show phytoplankton and phosphorus sensitivity to flow and water temperature conditions: i) reference simulation used throughout the manuscript; ii) constant low-flow in the Loire at S1 forced at 200 $m^3$ $s^{-1}$ and forced at 0.1 $m^3$ $s^{-1}$ in all other streams; iii) constant high-flow in the Loire at S1 forced at 1000 $m^3$ $s^{-1}$ and forced at 0.1 $m^3$ $s^{-1}$ in all other streams; iv) constant water temperature simulation, T = 13.7°C in all streams at all time. Phytoplankton development was much more affected by shorter travel times than by colder water temperature. P availability played a major role, and SRP exhaustion was reached 2.5 days after the starting date from S1 for all simulations except for the high-flow simulation where no P limitation was simulated because travel time from S1 to S2 was not long enough.